# Renewed Public Health Threat from Emerging Lyssaviruses

**DOI:** 10.3390/v13091769

**Published:** 2021-09-04

**Authors:** Anthony R. Fooks, Rebecca Shipley, Wanda Markotter, Noël Tordo, Conrad M. Freuling, Thomas Müller, Lorraine M. McElhinney, Ashley C. Banyard, Charles E. Rupprecht

**Affiliations:** 1Animal and Plant Health Agency (APHA), WHO Collaborating Centre (Rabies)/OIE Reference Laboratory (Rabies), Weybridge KT15 3NB, UK; rebecca.shipley@apha.gov.uk (R.S.); Lorraine.McElhinney@apha.gov.uk (L.M.M.); ashley.banyard@apha.gov.uk (A.C.B.); 2Centre for Viral Zoonoses, Department of Medical Virology, Faculty of Health Sciences, University of Pretoria, Pretoria 0001, South Africa; wanda.markotter@up.ac.za; 3Institut Pasteur de Guinée, Route de Donka, Conakry BP 4416, Guinea; ntordo@pasteur.fr; 4Friedrich-Loeffler-Institute (FLI), WHO Collaborating Centre for Rabies Surveillance and Research (Rabies)/OIE Reference Laboratory, 17493 Greifswald-Insel Riems, Germany; conrad.freuling@fli.de (C.M.F.); Thomas.Mueller@fli.de (T.M.); 5LYSSA LLC, Cumming, GA 30044, USA; charles_rupprecht@yahoo.com

**Keywords:** rabies, lyssavirus, bats, emerging, novel, encephalitis, prophylaxis, zoonoses

## Abstract

Pathogen discovery contributes to our knowledge of bat-borne viruses and is linked to the heightened interest globally in bats as recognised reservoirs of zoonotic agents. The transmission of lyssaviruses from bats-to-humans, domestic animals, or other wildlife species is uncommon, but interest in these pathogens remains due to their ability to cause an acute, progressive, invariably fatal encephalitis in humans. Consequently, the detection and characterisation of bat lyssaviruses continues to expand our knowledge of their phylogroup definition, viral diversity, host species association, geographical distribution, evolution, mechanisms for perpetuation, and the potential routes of transmission. Although the opportunity for lyssavirus cross-species transmission seems rare, adaptation in a new host and the possibility of onward transmission to humans requires continued investigation. Considering the limited efficacy of available rabies biologicals it is important to further our understanding of protective immunity to minimize the threat from these pathogens to public health. Hence, in addition to increased surveillance, the development of a niche pan-lyssavirus vaccine or therapeutic biologics for post-exposure prophylaxis for use against genetically divergent lyssaviruses should be an international priority as these emerging lyssaviruses remain a concern for global public health.

## 1. Introduction

Virus ‘hunting’ has stimulated the detection and characterisation of new lyssaviruses, most detected in chiropteran hosts [1] (1). Although cross-species transmission (CST) of lyssaviruses from bats-to-humans or any other mammalian species is rare [2,3,4,5,6] by comparison to the burden of canine rabies [7]. Interest in these RNA viruses remains high. All lyssaviruses cause clinical rabies and not a ‘rabies-like’ disease. Accurate estimates of human infection caused by lyssaviruses are imprecise, because of inadequate laboratory-based surveillance systems, mainly across Africa and Asia [8]. As is clear from the current COVID-19 pandemic, a more thorough understanding of bat ecology, immunology, and pathobiology will play an ever-increasing role in advancing our knowledge of the risks related to bat-transmitted diseases [9]. The paucity of relevant data, however, on bat populations, their distribution, relative abundance, and the sporadic interactions between bat populations and other taxa, such as carnivores, demonstrates that the role of bats and the epizootiology of host-virus dynamics are unclear. The extreme severity of disease caused by lyssaviruses means that the opportunity for CST is of major significance to human and animal populations. For rabies virus (RABV), most CST events are considered a dead-end infection, with infrequent virus transmission from an infected host, such as rabid vampire bat infections of humans or livestock [8,10]. In contrast, viral host switching events can result in sustained onward transmission to a naive host [11]. Host switching of RABV from bats appear to be more frequent in the Americas [12,13,14], whilst events involving other Old World (Africa, Europe, and Asia, or Afro-Eurasia) lyssaviruses appear to be rare [4,15]. There are multiple factors, including ecological and virological impacts, involved in CST and eventual host switching events [16,17]. In either case, endeavours to identify specific amino acid substitutions facilitating viral adaptation to new host species have been, for the most part, unsuccessful [18].

## 2. The Increasing Diversity of Lyssaviruses

Lyssaviruses are classified in the family *Rhabdoviridae* and the order *Mononegavirales*. Within the *Lyssavirus* genus, there are 17 different viral species recognized by the International Committee on Taxonomy of Viruses [19]. Classified as separate entities according to their genomic sequences, these include: *Aravan lyssavirus* (ARAV); *Australian bat lyssavirus* (ABLV); *Bokeloh bat lyssavirus* (BBLV); *Duvenhage lyssavirus* (DUVV); *European bat 1 lyssavirus* (EBLV-1); *European bat 2 lyssavirus* (EBLV-2); *Gannoruwa bat lyssavirus* (GBLV); *Ikoma lyssavirus* (IKOV); *Irkut lyssavirus* (IRKV); *Khujand lyssavirus* (KHUV); *Lagos bat lyssavirus* (LBV); *Lleida bat lyssavirus* (LLEBV); *Mokola lyssavirus* (MOKV); *Rabies lyssavirus* (RABV); *Shimoni bat lyssavirus* (SHIBV); *Taiwan bat lyssavirus* (TWBLV); *West Caucasian bat lyssavirus* (WCBV). One further novel lyssavirus, *Kotalahti bat lyssavirus* (KBLV), which was detected in a Brandt’s bat (*Myotis brandtii*) in Finland [20,21], and two sequences from a potentially novel lyssavirus, provisionally named Matlo bat lyssavirus (MBLV) [22,23], have been reported but remain as tentative species until fully characterised. Besides taxonomic classification into species, these viruses may also be grouped, according to genetics, using the nucleoprotein gene, phylogenetic, and antigenic data, into phylogroups [24]. At least two phylogroups can be identified by topological analysis of the phylogeny (Figure 1). The phylogroup I phylogenetic tree divides into two major branches, one of which is composed of the Palearctic and Indo-Malay regions (ARAV, BBLV, EBLV-2, KBLV, ABLV, GBLV, and KHUV); and another composed of EBLV-1, DUVV, IRKV, and TWBLV. Phylogroup II is composed of the African lyssaviruses, LBV, MOKVm and SHIBV. The most genetically divergent lyssaviruses have been tentatively classified within a dispersed phylogroup III, subdivided in two branches gathering viruses isolated from Europe to Africa, consisting of species, WCBV-MBLV and IKOV-LLEBV, respectively. In terms of sequence identity, viruses classified into phylogroup II exhibit the highest average nucleotide sequence similarity of the glycoprotein gene at 71.5%, whilst viruses classified into phylogroup I exhibit an average nucleotide sequence similarity of 70.3%. Viruses tentatively classified into phylogroup III exhibit an average nucleotide sequence similarity of 58.2%, indicating divergence of these viruses. In relation to viruses classified into phylogroup I and II, phylogroup III viruses exhibit an average nucleotide sequence similarity of 54.1% and 54.5%, respectively. With similarity to the evolutionary and genetic data (Figure 1), the antigenic analyses have identified several antigenic groups, referred to as phylogroups (Figure 1). Differentiation into these phylogroups predicts the degree of cross protection afforded by available rabies vaccines based on RABV. Though immunological responses to vaccines differ between individuals, it is widely accepted that the conservative virus neutralising antibody (VNA) level equal to or greater than 0.5 international units (IU/mL) positively correlates with seroconversion after vaccination against RABV [25]. Besides RABV, licensed rabies vaccines appear to largely confer protective immunity against phylogroup I lyssaviruses [26,27], with the least potential against IRKV [26,28]. Altogether, the level of VNA required for protection is undefined for non-RABV lyssaviruses [29,30].

For more divergent lyssaviruses, in vivo vaccination-challenge experiments have shown that the VNA response generated from RABV vaccines is inadequate to protect against challenge [26,31,32]. Phylogroup II lyssaviruses include LBV (all lineages, A-D), MOKV and SHIBV. Three further lyssaviruses, IKOV, LLEBV, and WCBV represent the most genetically and antigenically divergent lyssaviruses and these have been tentatively classified within a further phylogroup, phylogroup III (Figure 2). However, there appears to be limited cross neutralisation among these three viruses, and as such, they are unlikely to be antigenically categorised together in the same phylogroup. Phylogroup I viruses exhibit an average distance of 2 antigenic units (AU) to the SN strain, derived from the SAD B19 cDNA clone of RABV, equivalent to only a 4-fold difference in a VNA titre across multiple lyssavirus species [33]. In contrast, the highly genetically and antigenically divergent phylogroup III lyssaviruses demonstrate an average antigenic distance of 10 AU to the SN vaccine strain of RABV, equivalent to a 1024-fold difference in VNA titre (Figure 2).

Lyssaviruses have a distinctive epidemiology through their evolutionary association with bats [12]. RABV is present globally, being reported in carnivores and multiple other mammalian species across the New World, within multiple bat taxa. Whilst canine rabies has been largely eliminated in the Americas, curiously, of the 17 recognized lyssaviruses, only RABV has been reported in the New World (representing the Americas), where it is also identified within multiple bat taxa [8]. From a host-pathogen perspective, bat RABV in the New World has been associated with over 40 different species of insectivorous, hematophagous and frugivorous bats, including reservoirs among common genera, such as *Desmodus*, *Eptesicus*, *Lasiurus*, *Lasionycteris*, *Myotis*, *Perimyotis*, and *Tadarida* species, among others [2,5]. Hence, New World bat populations represent an omnipresent source of RABV infection for which elimination options remain unavailable. In contrast to the situation across the Americas, RABV has never been detected in bats in the Old World, yet the virus cycles among carnivores across much of the Old World [11,15,34]. This species-associated virus-host relationship of RABV has led to the suggestion of host restriction or co-evolution of pathogens with certain bat species (Table 1) [2,8]. However, there are partial exceptions to the rule. Detection of Old World lyssaviruses have been sporadic and opportunistic, complicating any inference on host species association and accurate distribution. For example, it remains enigmatic for WCBV and LLEBV appearing to occupy one and the same ecological niche, the Schreibers’ long-fingered bat (*Miniopterus schreibersii*) in Europe, although detection of both viruses has been relatively rare. ABLV and LBV have been detected in different species of bats in Australia and Africa, respectively, with LBV demonstrating a higher genetic diversity than any other of the known Old World bat lyssaviruses [35,36].

The relative proportion of the bat population able to transmit a lyssavirus to other host species is considered less than 1% (data not shown). Consequently, only a small number of human rabies cases caused by Old World lyssaviruses have been recognized [37]. Such paucity of data on Old World lyssaviruses may reflect fewer cases of exposure in humans than there are with RABV in the New World across human and animal populations. However, the basis for this observation remains enigmatic.

## 3. Are *Miniopterus* Species a Source of Genetically-Divergent Lyssaviruses?

More recently, there have been an increased number of reported cases and subsequently isolations of these most divergent and rare lyssaviruses. Retrospective characterization of laboratory samples from rabid animals indicated that such cases previously reported as rabies positive were due to Old World lyssaviruses [37]. Where surveillance efforts are implemented, increased reports of known lyssaviruses will ultimately follow [22]. Historically, mass mortalities have been reported in colonies of *Miniopterus schreibersii* throughout Spain, France, and Portugal [39]. In Spain, these events prompted investigations into the potential cause of the mortalities, although no causal relationship between pathogens and the fatalities were established. This bat species has also been associated with the proposed highly divergent phylogroup III lyssaviruses (Table 1). In the past three years, divergent lyssaviruses have been described as exemplified below, not only broadening the potential lyssavirus species list but also expanding the geographical distribution of previous known species.

## 4. Lleida Bat Lyssavirus in France

Most cases of bat rabies in Europe are associated with EBLV-1 [40,41]. In Spain, LLEBV was first detected in 2011, only the second phylogroup III lyssavirus from the continent, after WCBV [32,42]. With the turn of the century, a passive surveillance network was initiated in France, with the objective of detecting lyssaviruses among indigenous bat species. In 2017, a dead adult male *M. schreibersii* was diagnosed as positive for lyssavirus antigen and molecular testing characterized the sample as LLEBV, more than 700 km from the original report of LLEBV in Spain [43]. The detection of one of the most genetically divergent lyssavirus species, LLEBV, in a dead *M. schreibersii* bat in France stimulated further interest in these highly divergent lyssaviruses. Brain samples were found positive for both viral antigens and nucleic acids and a virus isolate was recovered. The genome sequence had 99.7% nucleotide identity with the Spanish LLEBV [44,45].

## 5. West Caucasian Bat Virus in Italy

In 2020, rabies was confirmed in a two-year-old cat in the Arezzo region. The cat died four days after the demonstration of clinical disease. Diagnostic evaluation by antigen and PCR testing confirmed the presence of a lyssavirus. Sequencing of the virus from the cat showed 98.5% homology with WCBV, suggesting a nearby bat colony may have been the source of the virus. WCBV has previously been detected in *M. schreibersi* in the Caucasus Mountains of South Eastern Europe [42] and cross-reactive antibodies against WCBV were detected in *M. schreibersi* in Kenya [38].

## 6. Matlo Bat Lyssavirus in The Republic of South Africa

During an enhanced survey in insectivorous bats in South Africa between 2003 and 2018, a new lyssavirus, MBLV, was identified [22,23] in the brain of two *Miniopterus natalensis* bats. This lyssavirus is 78.9% related phylogenetically to WCBV (20) (Figure 1). It is supposed that the greatest diversity of lyssaviruses is associated principally with bats on the African continent [15]. Intrinsically, in terms of bat host associations all recent detections of highly divergent bat lyssaviruses were associated with the *Miniopterus* bat genera with more than 30 species named so far, which is omnipresent in the Old World.

## 7. Bat Host Associations

It is notable that all the recent detections of highly divergent bat lyssaviruses in bats were associated with the *Miniopterus* bat genera. These bats have a broad range, including much of southern Europe, Africa, southern and south-eastern Asia, northern and eastern Australia, as well as the Melanesian Islands [46]. Previous detections of bat lyssaviruses indicated a clear geographical separation of Old World lyssaviruses. The recent detection of WCBV in Italy highlights the broad geographical range of this virus in *Miniopterus*. Moreover, MBLV is a likely sub-lineage of WCBV, expanding the range of this lyssavirus species into Africa. With the isolation of both WCBV and LLEBV in *M. schreibersii,* questions of basic host susceptibility arise from more than one virus being associated with the same species. With the evidence from these more recent lyssavirus notifications, there is a suggestion for both geographical distribution as well as a compartmentalised co-evolution of the bat species with association of a specific lyssavirus species. Regarding the latter, in fact, *Miniopterus* species are difficult to distinguish by external morphological features. This is particularly relevant in tropical regions of Africa where it is not yet possible to propose a clear taxonomy to prevailing species [47,48].

## 8. Availability of Vaccines for Bat Lyssaviruses

Continual recognition of novel emerging lyssaviruses warrants assessment of vaccine-derived cross reactivity. Modern human rabies vaccines and biologics have been available for decades. Following vaccination, a viral neutralising antibody titre above a defined threshold is considered a surrogate of efficacy for animal vaccination and also for humans at occupational risk of exposure to RABV [25]. However, this arbitrary cut off for a serological VNA titre is poorly defined for the other lyssaviruses. Certainly, reported variable efficacy of the WHO standard immune globulins and human sera from vaccinated personnel against more divergent phylogroup I viruses, such as DUVV and EBLV-1, and phylogroup II and III viruses suggests there is a cut-off threshold for antigenic relatedness for which human rabies vaccines will be ineffective [26,29,30,31]. Consequently, discovery of novel viruses warrants investigation on the safety, immunogenicity and efficacy of existing biologics, including the effectiveness of rabies immune globulin [25]. Detection of highly divergent lyssaviruses, LLEBV and WCBV in Europe and the putative MBLV in the Republic of South Africa, poses an increasing risk to human and animal populations alike, particularly if CST cases occur, especially as licensed biologics that have efficacy against these lyssavirus species do not exist. There are various avenues in how to overcome this limitation [49]. Current rabies vaccines are safe and, when administered properly, they are highly effective against RABV but lack efficacy against the highly divergent lyssavirus, particularly those lyssavirus species in phylogroup III. The surface glycoprotein G being the main immunogen for lyssaviruses, including a chimeric G protein RABV-MOKV or RABV-EBLV1 has shown that it is possible to enlarge the cross-neutralisation spectrum within the antigenic group I and between antigenic groups I and II, in the perspective of an anti-lyssavirus vaccine [50,51,52,53]. Therefore, new prototype vaccine candidates must have cost benefits, especially with broad immunologic and protective efficacy, as opposed to currently available vaccines, so these prototype vaccines will eventually enter clinical trials and become available on the global market [44,52].

## 9. Emerging Lyssaviruses

Although the number of reported historical human cases caused by a divergent lyssavirus is greater than 15 cases worldwide, this number is considered an underestimate due to underreporting and lower levels of typing of human rabies cases in the Old World, particularly in Africa. As long as biologics against these viruses are unavailable, reduction and prevention of exposure is the only available risk mitigating measure. Given the ongoing COVID-19 pandemic, there is an increased interest in zoonotic diseases and the availability of laboratory-based surveillance in support of national wildlife surveillance programmes. The development of surveillance programmes for detecting pathogens in wildlife offers an opportunity for future research on other viruses that may co-infect bats along with lyssaviruses and other emerging pathogens. Increased laboratory-based surveillance will support the discovery of novel lyssaviruses and will enhance knowledge on the distribution of recognised lyssaviruses. The increased use of pan-lyssavirus molecular tools and sequencing in routine diagnostic laboratories, as well as reduced reliance on RABV specific reagents, will also result in enhanced detection of non-RABV lyssaviruses in the future. Expanding this capacity worldwide may fill knowledge gaps and support the international health sectors in becoming more vigilant in recognising the risk of lyssaviruses to animal and human health.

## 10. Conclusions

In conclusion, further research under a ‘One Health’ programme is required to understand: the intraspecific mechanisms of perpetuation of viral pathogens within bat populations; the zoonotic potential of these viral pathogens detected in bats; the altered regional distribution of reservoirs such as bats, particularly as impacted by anthropological changes, habitat fragmentation and climate change; the risk of CST events that will impact both human and animal health.

## Figures and Tables

**Figure 1 viruses-13-01769-f001:**
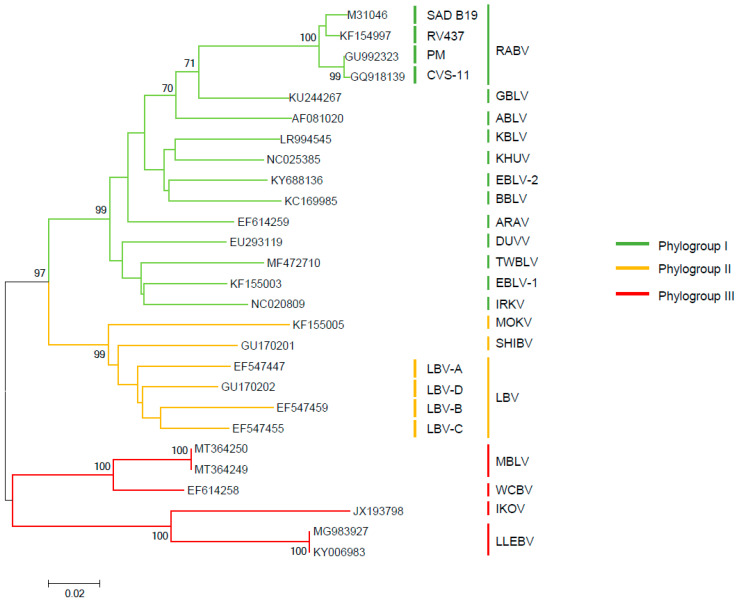
Phylogenetic reconstruction by inference of nucleoprotein gene sequences of lyssaviruses using the Neighbour-Joining method. The percentage of replicate trees in which the associated taxa clustered together in the bootstrap test (1000 replicates) are shown next to the branches. The GenBank accession numbers are indicated for each sequence. The evolutionary distances were computed using the Maximum Composite Likelihood method and are in the units of the number of base substitutions per site. Phylogenetic tree was generated in MEGA 6.

**Figure 2 viruses-13-01769-f002:**
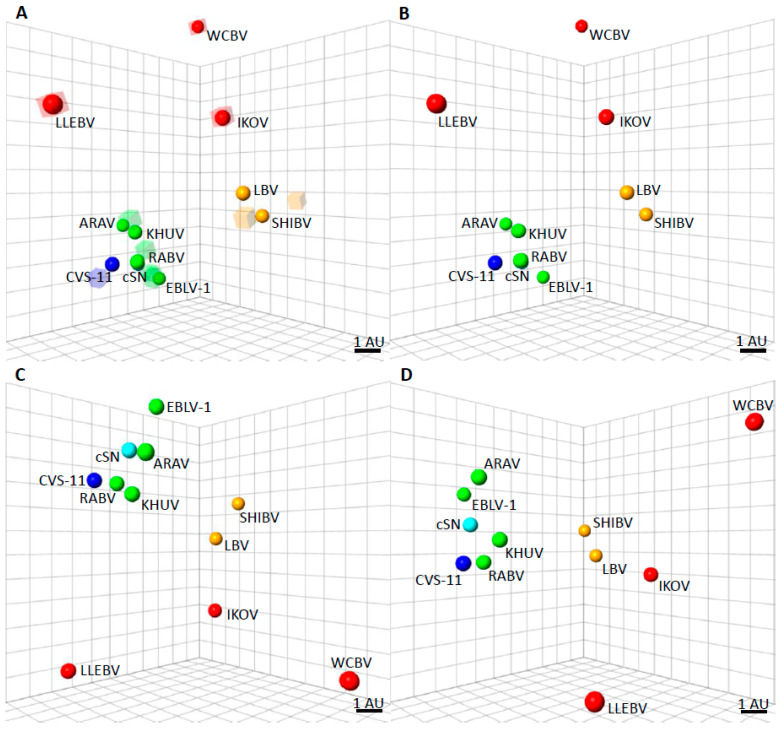
Antigenic cartography maps to show the antigenic distances of the lyssaviruses. (**A**) Three-dimensional antigenic map showing the antigenic relationship between lyssaviruses. Viruses (spheres) and, sera (translucent coloured boxes) are positioned such that the distance from each serum to each virus is determined by the neutralisation titre. Multidimensional scaling is used to position both sera and viruses relative to each other, so orientation of the map within the axes is free. The scale bar represents 1 AU (antigenic unit), equivalent to a two-fold dilution in antibody titre. Phylogroup I lyssaviruses are coloured green, CVS-11 coloured dark blue [Challenge virus standard-11 strain of RABV, used routinely in diagnostic assays], cSN [cDNA clone of the SN strain of RABV derived from the RABV strain, SAD B19] coloured light blue, Phylogroup II lyssaviruses coloured orange, and Phylogroup III lyssaviruses coloured red. (**B**) Antigenic map with sera removed for clarity. (**C**) Antigenic map, rotated to a different orientation and sera removed for clarity. (**D**) Antigenic map, rotated to a different orientation and sera removed for clarity.

**Table 1 viruses-13-01769-t001:** 21st century occurrence of the proposed ‘phylogroup III’ lyssavirus species.

Lyssavirus Species	Mammalian Isolate	Reservoir Species Associated with Lyssavirus Infection	Year of Isolation	Countries Reporting ‘Phylogroup III’ Lyssaviruses	Region
West Caucasian bat lyssavirus (WCBV)	Schreibers’ long-fingered bat (also known as the common bent-wing bat)	*M. schreibersii*	2002	Russian Federation	Eurasia
Ikoma lyssavirus (IKOV)	African civet (*Civettictis civetta*)	* Not known	2009	Tanzania	East Africa
Lleida bat lyssavirus (LLEBV)	Schreibers’ long-fingered bat	*M. schreibersii*	2011	Spain	Western Europe
Lleida bat lyssavirus (LLEBV)	Schreibers’ long-fingered bat	*M. schreibersii*	2017	France	Western Europe
West Caucasian bat lyssavirus (WCBV)	Domestic cat (*Felis catus*)	Suspected *M. schreibersii* colonies near the house where the cat resided.	2020	Italy	Western Europe
Matlo bat lyssavirus (MBLV)	Natal long-fingered bat	*M. natalensis*	2015–2016	Republic of South Africa	Southern Africa

* Virus neutralizing antibodies detected from *Miniopterus* species bats in Kenya reported during 2006–2007 [38].

## Data Availability

The data presented in this study are available on request from the corresponding author.

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
