# Peer review of "Renewed Public Health Threat from Emerging Lyssaviruses"

_viruses, 2021, doi:10.3390/v13091769_

Round 1

Reviewer 1 Report

The manuscript described the concern of public health threat from newly emerging lyssaviruses isolated from bats. Recent lyssaviruses represented highly divergent and immune escape against commercial rabies vaccines and biologics. If CST cases occur like COVID-19 outbreak, human and animal health could be problematic. Therefore, new vaccine candidate with broad protective efficacy should be developed. Ongoing surveillance programs for wild life by COVID-19 outbreak could support the discovery of new lyssaviruses and the epidemiologic information of these viruses. This manuscript should be helpful for understanding of recent bat lyssaviruses but some points should be additionally explained as below.

Major revision

  1. A recent published review paper is highly similar to this paper. More comprehensive information was already described in a previous report. The novelty of this paper should be explained (doi:10.3390/tropicalmed4010031).
  2. You mentioned that lyssaviruses (IKOV, LLEBV, WCBV) within phylogroup III represented higher genetically and antigenically divergence. In this manuscript, the comparative information of variability was not described between phylogroup I, II and phylogroup III. Additional information of sequence variability should be described proving increased variability of recent lyssaviruses based on sequence identity rates or presence of genotypes within each species.
  3. Before 21th century, limited information of lyssaviruses was described in the manuscript. Information of other lyssaviruses before 20th century should be added in table.
  4. CVS-11 strain should be additionally explained in Figure 1b.
  5. Definition of new world and old world should be explained
  6. Conclusion should be needed.

Minor revision.

  1. Typing mistakes are identified. All manuscript should be checked. For example, table title and ‘(WCBV}’ in table.
  2. Increased letter size were identified in line 159, 189 and 190.

Author Response

We wish to thank reviewer #1 for their constructive comments. Please see the attached rebuttal.

Reviewer 2 Report

This article is written well and provides the latest information on lyssaviruses, including their genetic and antigenic properties and ecology such as their hosts and geographical distributions. This also emphasizes the needs of pan-lyssavirus vaccine by addressing previous findings on predicted cross protection provided by current rabies vaccines.

I have only a few comments as follows:

  • Table: There is an error on the title.
  • Section 6 (lines 189-190): Font sizes in reference numbers are different from those in other parts.
  • Section 8 (lines 211-259):This section consists of only one paragraph, but it seems to me that it contains various topics. Please change the construction of this section so that readers can follow the topics easily.

Author Response

We wish to thank reviewer #2 for their constructive comments. Please see the attached rebuttal.

Round 2

Reviewer 1 Report

I acknowledge all the author's replies. The authors have addressed the comments and I recommend acceptance of the manuscript. 

Author Response

I acknowledge all the author's replies. The authors have addressed the comments and I recommend acceptance of the manuscript. 

We wish to thank Reviewer #1 and have undertaken a check of all works in the body of the text. Completed.